# An Overview of Pancreatic Neuroendocrine Tumors and an Update on Endoscopic Techniques for Their Management

Osama O. Elkelany [1], Fred G. Karaisz [2], Benjamin Davies [3] and Somashekar G. Krishna [2,*]

1    Department of Internal Medicine, The Ohio State University Wexner Medical Center,
     Columbus, OH 43210, USA
2    Division of Gastroenterology, Hepatology and Nutrition, Department of Internal Medicine, The Ohio State
     University Wexner Medical Center, Columbus, OH 43210, USA
3    College of Medicine, The Ohio State University, Columbus, OH 43201, USA
*    Correspondence: somashekar.krishna@osumc.edu

**Abstract:** The growing importance of advanced endoscopy in the diagnosis and treatment of pancreatic neuroendocrine neoplasms (PanNETs) necessitates a comprehensive understanding of various biochemical markers, genetic testing methods, radiological techniques, and treatment approaches that encompass multiple disciplines within and beyond gastrointestinal oncology. This review aims to highlight key aspects of these topics, with a specific focus on emerging EUS-guided procedures for the management of PanNETs.

**Keywords:** pancreatic neuroendocrine tumor (PanNET); endoscopic-ultrasound-guided ethanol ablation (EUS-EA); endoscopic-ultrasound-guided radiofrequency ablation (EUS-RFA)

## 1. Introduction

Pancreatic neuroendocrine neoplasms (PanNENs) include pancreatic neuroendocrine tumors (PanNETs) and pancreatic neuroendocrine carcinomas (PanNECs) [1,2]. PanNETs are well-differentiated neoplasms of the pancreas with a diverse pathophysiology underscoring the complex mechanisms of the gastrointestinal hormones that are often involved. PanNETs, constituting 1–2% of pancreatic cancers, exhibit an incidence of approximately 5 in 100,000 individuals. The growing incidence can be attributed in part to the heightened rate of incidental detection [3–5]. These neoplasms can develop sporadically or as a manifestation of a familial syndrome [6–8].

From a clinical standpoint, PanNETs are broadly classified into two groups: functional and nonfunctional. Functional PanNETs, which comprise 34.5% of all PanNETs [9], exhibit the excessive secretion of various biologically active peptides, such as insulin or glucagon, leading to the manifestation of diverse syndromes. On the other hand, nonfunctional PanNETs lack the oversecretion of such peptides but share similar histological and pathological characteristics with functional PanNETs [6]. The treatment goals for functional PanNETs involve eliminating neuroendocrine tumor cells to halt hormonal hypersecretion and prevent malignant spread [10]. In contrast, the management of nonfunctional PanNETs is more complex, focusing on predicting and impeding tumor growth and progression [11]. Innovative endoscopic approaches like EUS-guided radiofrequency ablation (RFA) and EUS-guided fine needle injection (FNI) of a chemoablative agent hold promise as effective alternatives to surgical pancreatectomies [12]. With the expanding role of endoscopists in the diagnosis and management of PanNETs, it is imperative to possess a comprehensive understanding of the available tools and approaches to effectively address these conditions.

## 2. Diagnosis of Functional PanNETs

### *2.1. Insulinomas*

Insulinomas are the most common functional PanNETs, accounting for 20.9% of cases [9]. Typically, insulinomas are benign and well-differentiated NETs; however, approximately 5.8% of insulinomas are malignant [13,14]. While insulinomas are usually sporadic, around 4–5% of patients with insulinomas have multiple endocrine neoplasia type 1 (MEN1) [15]. The gold standard for diagnosing insulinomas is measuring insulin levels after a 72 h fasting test, which demonstrates close to 100% sensitivity and specificity [16]. Once organic hyperinsulinism is confirmed in symptomatic patients (as outlined in Table 1), imaging is necessary to locate the tumor for surgical management. Various imaging modalities can be employed, including ultrasound (US), computed tomography (CT), magnetic resonance imaging (MRI), endoscopic ultrasound (EUS) with fine needle aspiration (FNA), arteriography, intra-arterial stimulation with venous sampling (ASVS), and somatostatin receptor scintigraphy [17]. 68Ga-DOTATATE positron emission tomography (PET/CT) and Fluorine-18-L-dihydroxyphenylalanine (18-F-DOPA) PET, which take advantage of PanNET's propensity for decarboxylate amine precursors, have been found to be more sensitive than CT or MRI in identifying insulinomas or beta cell hyperplasia and are used when conventional imaging modalities yield inconclusive results [15,18]. Regarding the additional benefits of using EUS, a meta-analysis conducted by James et al. revealed that the implementation of EUS was associated with a higher detection rate of PanNETs even after the utilization of CT and MRI imaging. This contributed to an overall increase in PNET detection of more than 25% [19]. In a retrospective study conducted by Pais et al., the sensitivity of EUS-FNA in diagnosing PanNETs was found to be 87%. Notably, this sensitivity remained consistent in both functional and nonfunctional PanNETs. These findings further demonstrate that EUS can be a valuable tool to aid in the diagnosis of PanNETs [20].

### *2.2. Gastrinomas*

The second most common functional PanNETs are gastrinomas (8.2%) [9]. Gastrinomas secrete gastrin and cause Zollinger–Ellison syndrome (ZES) [21]. This syndrome is characterized by hypersecretion of gastric acid leading to peptic ulcer disease and gastroesophageal reflux disease. It is estimated that 25% of gastrinomas occur in patients with MEN1 [22]. A more than one-thousand-fold increase in gastrin levels can be diagnostic [23]. If gastrin levels are only moderately elevated, a secretin test is required, during which gastrin levels are measured after intravenous administration of a secretin bolus [24]. However, gastrin levels can be elevated in patients with atrophic gastritis or in patients receiving proton pump inhibitor therapy [25]. As a result, the diagnostic criteria for ZES are not commonly fulfilled, and greater emphasis is placed on imaging modalities to detect the presence of an intra-abdominal tumor [26].

In a prospective study involving 80 patients with gastrinomas, somatostatin receptor scintigraphy (Octreotide scan) alone yielded a gastrinoma detection rate of 58%, while conventional imaging modalities such as ultrasound (US), computed tomography (CT), magnetic resonance imaging (MRI), and angiography had a detection rate ranging from 9% to 31% [27]. This scintigraphy test utilizes an indium-111 radiolabel that binds to somatostatin receptors, providing an overall sensitivity of 75–100% for detecting PanNETs [28,29]. To enhance its sensitivity, this technique can be combined with single-positron emission computed tomography (SPECT) [30]. More recently, somatostatin receptor PET imaging using three FDA-approved radiotracers, namely gallium-68-dodecanetetraacetic acid, tyrosine-3-octreotate (68-Ga-DOTATATE), 68-Ga-DOTATOC, and 64-Cu-DOTATAT, has emerged as superior to the traditional octreotide scan. These radiotracers have demonstrated high sensitivity, shorter imaging duration, and lower radiation doses. In PanNETs with lower somatostatin receptor expression, imaging with (18)F-FDG PET is preferred [31].

**Table 1.** Characteristics of PanNETs.

| PanNET Type | Associated Biomark-ers/Hormones | Associated Genetic Syndromes | Clinical Manifestations | Diagnosis | Imaging |
|---|---|---|---|---|---|
| **Insulinoma** | Insulin Pro-insulin C-Peptide Chromogranin A | MEN1 | Hypoglycemia | 72 h fasting test  Negative serum/urine toxicology for insulin secretagogues | Ultrasound/CT/MRI/EUS  Somatostatin receptor scintigraphy (octreotide scan)  Arteriography/intraarterial stimulation with venous sampling  18-F-DOPA PET  68Ga-DOTATATE PET/CT |
| **Gastrinoma** | Gastrin Chromogranin A | MEN1 | Zollinger–Ellison syndrome | Gastrin levels  Stomach pH  Secretin test | EUS  Somatostatin receptor scintigraphy (octreotide scan)  68Ga-DOTATATE PET/CT |
| **Glucagonoma** | Glucagon Chromogranin A Neuron-specific enolase | MEN1 | Migratory necrolytic erythema | Glucagon levels | Ultrasound/CT  Somatostatin receptor scintigraphy (octreotide scan) |
| **VIPoma** | Vasoactive intestinal polypeptide Gastrin Insulin | MEN1 | Secretory diarrhea | VIP levels  Stool osmolality  68Ga-DOTATATE PET/CT | 68Ga-DOTATATE PET/CT |
| **Somatostatinoma** | Somatostatin | MEN1 | Somatostatinoma syndrome | EUS-FNA | Ultrasound/CT/MRI |
| **Nonfunctional PanNETs** | Chromogranin A Polypeptide P | VHL | | | EUS |

While EUS demonstrates excellent sensitivity and specificity for detecting gastrinomas in the pancreas, its sensitivity may decrease when gastrinomas are localized in the duodenum. However, the high sensitivity of EUS in detecting small PanNETs (less than 2 cm) has led some experts to propose an annual EUS screening for asymptomatic patients with MEN1 [32].

*2.3. Glucagonomas, VIPomas, and Somatostatinomas*

For the diagnosis of glucagonomas, glucagon levels above 500 pg/mL are commonly observed [33]. However, it is important to note that elevated glucagon levels can also be seen in other conditions such as cirrhosis and diabetes. Therefore, the interpretation of elevated glucagon levels should be combined with the presence of typical glucagonoma syndrome symptoms, including weight loss, necrotic migratory erythema, and hypoalbuminemia [34].

In a study investigating a cohort of 1000 patients with various causes of diarrhea, elevated vasoactive intestinal peptide (VIP) levels were found to be 100% specific in diagnosing VIPomas [35]. This finding is particularly relevant as VIPomas typically present with symptoms such as watery diarrhea, hypokalemia, and achlorhydria. The study demonstrated the high specificity of VIP levels in aiding the diagnosis of VIPomas. It is worth noting that other peptides, including gastrin and insulinoma, can also be elevated in patients with VIPomas [36].

Somatostatinomas can produce a "stomatostatinoma syndrome", which is characterized by anemia, diabetes, diarrhea, and gallbladder disease [6]. Fasting plasma somatostatin level greater than 14 mol/L and CT of the abdomen are often diagnostic, as these tumors, like glucagonomas, are often large upon presentation [37].

## 3. Diagnosis of Nonfunctional PanNETs

Nonfunctional PanNETs typically do not cause symptoms until they reach a significant tumor burden or when complications arise from mass effect or metastatic disease. Despite their indolent nature, over 11% of nonfunctional PanNETs are diagnosed as distant metastases. The diagnosis of nonfunctional PanNETs involves a combination of biochemical, radiographic, and histologic evaluations [6,37–39].

*Neuroendocrine Biomarkers*

Neuroendocrine biomarkers play a role in diagnosing and surveilling and are applicable to both functional and nonfunctional PanNETs [40]. One of the most ubiquitously measured biomarkers is a glycoprotein produced by neuroendocrine cells known as chromogranin A (CgA). It has been shown that CgA can affect several elements in the tumor microenvironment, including endothelial cells and fibroblasts. These findings have also suggested that its abnormal secretion could play a role in tumor progression [41]. In the diagnosis of functional and nonfunctional PanNETs, CgA has been found to have a sensitivity of 66% and specificity of 95% [42]. CgA can also be false-positively elevated in several clinical conditions including inflammatory bowel disease, renal failure, liver failure, or pancreatitis. In addition, proton pump inhibitor (PPI) therapy as well steroids have also been shown to increase CgA levels [43,44], and this effect by PPIs can last up to 2 weeks [45]. While these circumstances may limit CgA's use as a diagnostic biomarker, evidence is favorable for its use as a prognostic factor for progression-free survival (PFS) and overall survival (OS) [42,44].

Neuron-specific enolase (NSE), a glycolytic enzyme expressed in neuroendocrine cells, is another biomarker that has demonstrated prognostic value in progression-free survival and overall survival. The expression of NSE occurs as a late event in the neural cell differentiation process [46]. Its utility as a biomarker is based on its expression in neural maturation and tumor proliferation. While NSE has a diagnostic sensitivity of 31% in comparison to CgA in PanNETs, elevated baseline NSE values similarly showed prognostication on progression-free survival and overall survival [47,48]. Furthermore, early decreases in CgA and NSE levels after treatment can serve as a prognostic marker in patients with advanced PanNETs. The combined use of CgA and NSE has been shown to be more accurate in predicting and prognosticating disease [49].

One of the more novel biomarkers involves measuring circulating neuroendocrine tumor transcripts (NETest). This test uses PCR to measure 51 different transcripts related to neuroendocrine tumors or associated with neoplastic behavior and was found to have a sensitivity of 80% and specificity of 94% for PanNETs in a study that investigated 206 patients with neuroendocrine tumors [50]. The genes tested include KRAS, RAF1, and APLP2, among others. NETest is also associated with disease progression and can potentially be a better predictor of disease than CgA [50,51].

## 4. Genetics of PanNETs

Inherited disorders associated with a relatively high incidence of PanNETs include multiple endocrine neoplasia type 1 (MEN1), von Hippel–Lindau disease, von Recklinghausen's disease (neurofibromatosis 1), and tuberous sclerosis.

Although most PanNETs occur as sporadic tumors, they can also arise in association with syndromes such as multiple endocrine neoplasia type 1 (MEN1) and von Hippel–Lindau (VHL). MEN1 results from a mutation of the MEN1 gene on chromosome 11q13, which acts as a tumor suppressor gene [52]. Approximately 21% of sporadic PanNETs have mutations in the MEN 1 gene [53]. These MEN1 mutations are more frequent in gastrinomas (37%), VIPomas (44%), and glucagonomas (67%), while less frequent in insulinomas and nonfunctioning PanNETs (8%) [54,55]. Up to 68% of sporadic PanNETs also exhibit a loss of heterozygosity at chromosome 11q13 [56]. A study carried out by Scarpa et al. investigated the genomic landscape of PanNETs [57]. They found that the base excision repair gene MUTYH was mutated, which affected the process of DNA damage repair. They also found that genes such as MEN1, SETD2, ARID1A, and MLL3, which are all involved in chromatin remodeling, were inactivated, leading to a dysregulation of the transcription process.

## 5. Staging and Grading

The eighth edition of the American Joint Committee on Cancer (AJCC) incorporated the classification criteria outlined by the European Neuroendocrine Tumor Society (mENETS) guidelines, as depicted in Tables 2 and 3 [58].

**Table 2.** AJCC 8th staging classification of PanNET [58].

| | |
|---|---|
| T1 | Tumor limited to the pancreas, <2 cm |
| T2 | Tumor limited to the pancreas, 2–4 cm |
| T3 | Tumor limited to the pancreas, >4 cm, or invading the duodenum or common bile duct |
| T4 | Tumor invades adjacent structures * |
| N0 | No regional lymph node metastasis |
| N1 | Regional lymph node metastasis |
| M0 | No distant metastasis |
| M1 | Distant metastasis |

* Adjacent structures that can be involved in PanNETs include the stomach, spleen, colon, adrenal gland, or the walls of large vessels such as the celiac axis or superior mesenteric artery.

**Table 3.** The prognostic staging of PanNETs as determined by combining the criteria of the 8th AJCC * and ENETS guidelines [58].

| Stage | T | N | M |
|-------|-----|-------|-----|
| I | T1 | N0 | M0 |
| II (A) | T2 | N0 | M0 |
| II (B) | T3 | N0 | M0 |
| III (A) | T4 | N0 | M0 |
| III (B) | Any T | N1 | M0 |
| IV | Any T | Any N | M1 |

* AJCC: American Joint Committee on Cancer; ENETS: European Neuroendocrine Tumors Society.

EUS-FNA is a commonly used diagnostic procedure for PanNETs. Multiple studies have reported a sensitivity ranging from 73.2% to 100% and a specificity ranging from 83.3% to 93% in the diagnosis of PanNETs [59–66]. The grading of PanNETs follows the pathological classification outlined by the World Health Organization (WHO). The prognostic evaluation of PanNETs is reliably determined by assessing the mitotic index and Ki-67 index [67]. The concordance of PanNET Ki-67 grading between EUS-FNA specimens and surgical specimens is well established and is approximately 77.5% [67,68]. The sensitivity of these markers for Grade 1 PanNETs was found to be 91.4%, while it was lower for Grade 2 and Grade 3 at 55.7% and 59.0%, respectively. This discrepancy is believed to be due to the intratumoral heterogeneity of Ki-67, prompting the recommendation to count more than 2000 cells to improve the ability of EUS-FNA in grading diagnosis [69].

For cystic PanNETs, EUS-FNA can also be used to conduct cyst fluid analysis, which typically yields low CEA and amylase levels [70]. A smear of the aspirate will show predominantly isolated cells that are uniformly round with plasmacytoid morphology. The nucleus is round with finely stippled chromatin. Immunostaining can also help with finalizing the diagnosis [71]. Neuroendocrine tumors express chromogranin. Beta-catenin is also expressed, typically in the cytoplasm [72,73]. Another noteworthy diagnostic tool includes EUS-guided needle-based confocal laser endomicroscopy (EUS-nCLE), which identifies cystic PanNETs by their typical trabecular network of cells (in clusters) [74].

## 6. Management of PanNETs—Minimally Invasive Approaches

Pancreatectomy is considered the primary treatment for PanNETs, and surgery is also recommended for cases involving the main pancreatic duct, bile duct, or lymph nodes. However, despite being the gold standard, surgery is associated with significant adverse events in both the short and long term. These complications include pancreatic fistula, with a reported occurrence of 45% after tumor enucleation and 57% after central pancreatectomy according to a recent systematic review. Additionally, delayed gastric emptying was observed in 5% of cases after tumor enucleation and 15% after central pancreatectomy, while postoperative hemorrhage occurred in 6% of cases. The in-hospital mortality rate was reported as 4% for both distal pancreatectomy and central pancreatectomy [75]. Consequently, less-invasive alternative interventions have been explored, such as ethanol ablation and radiofrequency ablation under EUS guidance (EUS-EA and EUS-RFA).

Functional PanNETs generally have a low risk of malignancy, and the primary goal of treatment is to control the hormonal hypersecretion responsible for symptoms by targeting and eradicating enough neuroendocrine tumor cells [76]. Therefore, complete tumor ablation is not always necessary. In the case of nonfunctional PanNETs, the management approach becomes more complex. Some studies support the resection of all nonfunctional PanNETs to prevent tumor growth and progression. However, other studies have adopted a nonoperative management strategy for asymptomatic nonfunctional PanNETs that are discovered incidentally and measure smaller than 2 cm [77–82].

The conservative approach has been considered a reasonable strategy, as most of the investigated tumors do not exhibit significant changes during surveillance. Based on existing data, the ENETS published guidelines recommending surveillance for patients with lesions smaller than 2 cm [15]. However, it is important to note that other studies have indicated that larger PanNET lesions, up to 3 cm in size, may not consistently correlate with tumor behavior, as some of these lesions are associated with a high tumor grade and lymph node metastases [83,84].

### 6.1. EUS-Guided Ethanol Ablation in PanNETs

EUS ethanol ablation involves the injection of 95% ethanol into the center of the tumor until a hyperechoic blush is observed, expanding within the tumor on ultrasound imaging [85]. Table 4 summarizes current evidence.

**Table 4.** Case series and prospective studies of EUS-EA for PanNETs (Reprinted/adapted from [86]).

| Reference | Year | No. of Patients | Mean Size of Tumor (mm) | Complete Response Rate (%) | Complications (%) | Recurrences (%) |
|---|---|---|---|---|---|---|
| Levy et al. [85] | 2012 | 5 | 15 | 100% | 0% | Not specified |
| Choi et al. [87] | 2018 | 33 | 11 | 60% | 12.1% | Not specified |
| Matsumoto et al. [88] | 2020 | 5 | 10.2 | 80% | 0% | 0% |
| Park et al. [89] | 2015 | 11 | 12.3 | 53.8% | 36% (3 cases of pancreatitis, 1 case of abdominal pain) | Not specified |
| Paik et al. [90] | 2016 | 8 | 15 | 75% | 12.5% (2 cases of abdominal pain, 1 case of limited fever) | 37.5% |
| Yang et al. [91] | 2015 | 4 | Not specified | 75% | 0% | Not specified |

### 6.2. EUS-Guided Radiofrequency Ablation for PanNETS

EUS-RFA devices are capable of generating high temperatures ranging from 60 to 100 degrees Celsius, which can be used to induce tissue injury, apoptosis, and coagulative necrosis. Two devices have been studied for this purpose.

One of the initial studies examining the efficacy of EUS-RFA in PanNETs involved 10 patients, including 3 with functional PanNETs and the remaining with nonfunctional PanNETs. The mean size of the tumors was 1.6 cm. Complete ablation was successfully achieved in all patients, but three cases of acute pancreatitis were reported as postprocedural complications. Among these, two patients required endoscopic drainage of fluid collections that developed subsequently. During a median follow-up period of 34 months, no recurrences were observed [92].

Several case reports are published that show the efficacy and safety of EUS-RFA being used to successfully PanNETs as listed in Table 5 [93–96].

**Table 5.** Case-series and prospective studies of EUS-RFA for PanNETs (Reprinted/Adapted from [86]).

| Reference | Year | No. of Patients | Mean Size of Tumor (mm) | Complete Response Rate (%) | Complications (%) | Recurrences (%) |
|---|---|---|---|---|---|---|
| Rossi et al. [92] | 2014 | 10 | 16 | 100% | 30% (acute pancreatitis) | 0% |
| Armellini et al. [93] | 2015 | 1 | 20 | 100% | 0% | 0% |
| Pai et al. [97] | 2015 | 2 | 27.5 | 100% | 0% | 0% |

**Table 5.** *Cont.*

| Reference | Year | No. of Patients | Mean Size of Tumor (mm) | Complete Response Rate (%) | Complications (%) | Recurrences (%) |
|---|---|---|---|---|---|---|
| Lakhtakia et al. [98] | 2016 | 3 | 17 | 100% | 0% | 0% |
| Waung et al. [95] | 2016 | 1 | 18 | 100% | 0% | 0% |
| Bas-Cutrina et al. [94] | 2017 | 1 | 10 | 100% | 0% | Not specified |
| Choi et al. [99] | 2018 | 8 | 19 | 75% | 25% (1 case of abdominal pain; 1 case of acute pancreatitis) | Not specified |
| Thosani et al. [100] | 2018 | 3 | Not specified | 100% | Not specified | Not specified |
| de Mussy et al. [96] | 2018 | 1 | 18 | 100% | 0% | 0% |
| Barthet et al. [101] | 2019 | 12 | 13.1 | 85% | 14% (1 acute pancreatitis, 1 pancreatic duct stenosis) | Not specified |
| Oleinikov et al. [102] | 2019 | 18 | 14.3 | 96% | 0% | 0% |
| de Nucci et al. [103] | 2020 | 10 | 14.5 | 100% | 20% (2 cases of abdominal pain) | 0% |
| Younis et al. [104] | 2022 | 1 | 8.9 | 66.7% | Not specified | Not specified |
| Marx et al. [105] | 2022 | 7 | Not specified | 85.7% | 42% (minor adverse events) | Not specified |
| Ferreira et al. [106] | 2022 | 29 | 14.4 | 73.3% | 10% (acute pancreatitis) | Not specified |

A larger case series conducted by Choi et al. included seven patients with nonfunctional PanNETs, with a mean size of 20 mm. The study reported a complete response rate of 71.4%. After the procedure, one case of pancreatitis and one case of abdominal pain were observed [99]. Another case series conducted by de Nucci et al. involved 10 patients with a total of 11 PanNETs. The mean size of the lesions was 14.5 mm. Complete response was achieved in 100% of patients at both the 6-month and 12-month follow-up. The RFA procedure did not result in any major complications, and mild abdominal pain occurred in two cases [103].

In a single-center study by Younis et al., seven patients with PanNETs, with a median size of 8.9 mm, were included. A complete response was achieved in 66.7% of patients with nonfunctional PanNETs [104].

In a multicenter study by Barthet et al., 12 patients underwent EUS-RFA for a total of 14 nonfunctional PanNETs. The mean size of the lesions was 13.4 mm. At a 3-year follow-up, a complete response was achieved in 85.7% of cases [107]. Another recent multicenter study conducted by Ferreira et al. included 29 patients with a total of 35 lesions. Among these, 13 were functional PanNETs (specifically insulinomas) and 10 were nonfunctional PanNETs. The mean size of the lesions was 14.4 mm. Technical success was achieved in 100% of the cases, and no serious adverse events were reported. Mild pancreatitis occurred in approximately 10% of the patients. At the 6-month follow-up, a significant response was observed in 73.3% of the 15 PanNETs assessed, with 46.6% achieving complete necrosis and 26.7% experiencing a size reduction of over 50% [106].

Marx et al. conducted a multicenter study on EUS-RFA for the management of functional PanNETs. 7 patients were included, and 6 out of 7 patients achieved a complete response at a median follow-up of 21 months [105]. Three patients experienced minor adverse events. There was one mortality in a patient (age of 97 years) with a postprocedure retrogastric collection; the patient opted for supportive care without interventions including drainage. Oleinikov et al. conducted a multicenter study involving 18 patients, out of which 7 had functional PanNETs (specifically insulinomas), while the remaining patients

had nonfunctional PanNETs. The mean size of the lesions was 14.3 mm, with a total of 27 lesions among the patients. A remarkable complete response rate of 96% was achieved, and no adverse events were reported following the procedure. During a mean follow-up period of 8.7 months, no recurrences were observed [102].

## 7. Discussion

EUS offers diagnostic and therapeutic tools that address the unique clinical challenge posed by PanNETs. These therapeutic options (EUS-RFA and EUS-EA) provide a potential alternative to morbid pancreatectomies. These early prospective studies have largely shown high technical success rates, low complication rates, and promising response rates. A noteworthy systematic review and meta-analysis of this topic investigated the effectiveness and safety profile of both EUS-RFA and EUS-EA in PanNETs [108]. Although the mean size of PanNETs in the EUS-RFA group was significantly higher at 16.4 mm compared to 12.2 mm in the EUS-EA group, the investigators found that the overall rate of clinical success after EUS-RFA and EUS-EA was 85.2% and 82.2%, respectively. Overall, there was no statistical difference observed between the two techniques in terms of clinical success.

As previously highlighted, the definition of clinical success varies for functional and nonfunctional lesions. For functional lesions, clinical success entails ablating a sufficient portion of the lesion to achieve symptom resolution during follow-up. For nonfunctioning lesions, clinical success is defined as complete ablation as observed on follow-up imaging using CT or EUS. Technical success rates between EUS-RFA and EUS-EA have been found to be similar: 94% and 96.7%, respectively [108]. Regarding adverse events, the rate was 14.1% for EUS-RFA and 11.5% for EUS-EA, and there was no significant difference between them. The most reported adverse event was pancreatitis, accounting for 50% of all reported adverse events, followed by abdominal pain at 45.5%. The authors in this meta-analysis found that the location of the PanNET lesion in the head or neck of the pancreas was the only significant positive predictor of clinical success in EUS-RFA. They also observed a positive trend indicating a higher rate of adverse events with increased ethanol use in EUS-EA, although statistical significance was not reached. Interestingly, the size of the lesions did not have a significant impact on clinical or technical success, or adverse events. This is in contrast to another preceding systematic review by Imperatore et al., which demonstrated that lesions smaller than 18 mm had a 97% positive predictive value for response to EUS-RFA [109].

Large, multicenter studies are needed to firmly establish EUS procedures as a reliable means of treatment for PanNETs. However, there are several obstacles to conducting such studies, including variability in defining clinical success and the relative rarity of PanNETs. Among these procedures, perhaps the most impressive outcomes have been observed in functional PanNETs, particularly insulinomas, treated with EUS-RFA. Several studies have reported a near-complete resolution of symptoms after treatment for insulinomas [101,102,106]. Notably, these studies have encountered few serious complications from these treatments. Longer-term follow-up is necessary, especially given the breadth of disease caused by various PanNETs; however, sustained response among nonfunctional PanNETs treated by EUS-RFA has been demonstrated up to 3 years following treatment [101,107].

## 8. Conclusions

PanNETs are uncommon tumors that exhibit diverse outcomes based on stage, grade, and clinical presentation. Over the past few decades, there has been a notable rise in the incidence of PanNETs, particularly in the early stages of the disease due to improved imaging technology and frequency. Current international guidelines recommend active surveillance for small, well-differentiated, asymptomatic, nonfunctioning PanNETs measuring 1.5 to 2 cm. However, even with surveillance, there remains a possibility of disease progression. There is a clear need for an alternative approach to standard surgical treatment in certain cases. This includes patients with low-grade PanNETs measuring less than 20 mm who

may not be suitable candidates for surgery or are at high risk during the perioperative period. Additionally, functional tumors that require the removal of hormone-secreting cells may benefit from a debulking procedure. In these scenarios, EUS-RFA emerges as a promising alternative. It offers high efficacy in tumor ablation, a low rate of adverse events, and notable advantages such as minimal invasiveness and the potential for repeat procedures when necessary.

**Author Contributions:** F.G.K. and O.O.E., writing, extensive literature search; B.D., writing; S.G.K., writing process and critical revision of the final manuscript. All authors have read and agreed to the published version of the manuscript.

**Funding:** This research received no external funding.

**Conflicts of Interest:** Krishna, S.G.—research grant support (investigator-initiated studies) from Mauna Kea Technologies, Paris, France, and Taewoong Medical, USA.

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
