# Peer review of "An Overview of Pancreatic Neuroendocrine Tumors and an Update on Endoscopic Techniques for Their Management"

_curroncol, doi:10.3390/curroncol30080549_

Round 1

Reviewer 1 Report

The review titled "An Overview of Pancreatic Neuroendocrine Tumors and an Update on Endoscopic Techniques in their Management" comprehensively discusses the biochemical markers, genetic testing, radiological techniques, and treatment approaches involved in pancreatic neuroendocrine tumors (PanNETs). The review is well-written and provides informative insights into the recent diagnostic procedures used for PanNETs. However, I would like to add the following points:

  • While this review provides an explanation of the genetics of PanNETs (Line: 152), it would be beneficial to include information about the specific mutations and aberrant signaling mechanisms that contribute to the progression of PanNETs. A recent report delves into the genomic landscape of PanNETs, offering a comprehensive outlook on the mutations involved (PMID: 28199314). It would be valuable to incorporate details regarding these genetic aberrations in PanNETs.

The review mentions the use of polymerase chain reaction (PCR) tests to measure 51 different transcripts related to neuroendocrine tumors or associated with neoplastic behavior, which aids in characterizing PanNETs (Line 146). It would greatly enhance the review to provide a list of these 51 genes that are utilized for this purpose.

 Please note that these additions will further enhance the understanding of PanNETs, genetic aberrations, and the diagnostic procedures discussed in the review.

Author Response

Attached below

Reviewer 2 Report

The authors have set out to review pNETs and the role of endoscopy (like EUS) in the management. The first half is devoted to background information on pNETs while the second half dives more into the role of EUS.

The manuscript is well written and does cover a fair amount of material.

Some comments/questions does arise from reviewing the manuscript.

1. Tables 2A and 2B are from the AJCC. Is there permission to reproduce this in the manuscript?

2. The authors have cited the systemic review and meta-analysis by Garg et al., published last year. I think the authors need to justify a bit how different their manuscript is from the meta-analysis (what sets this manuscript apart from what has already been published). I think if it is suppose to be a one-stop shop for pNETS then there could be a further dive into the topic and literature review.

3. There are a couple of studies that I thought should have been mentioned in this manuscript (James et al., Incremental benefit of preoperative EUS for the detection of pNETs. A meta-analysis.  Gastrointest Endosc. 2015 Apr;81(4):848-56.; and Pais EUS for pancreatic neuroendocrine tumors: a single-center, 11-year experience. Gastrointest Endosc. 2010 Jun;71(7):1185-93.). The endoscopists I know have cited these in our discussions it might be good to look at these to see if they fit your manuscript).

4. On page 1, line 25 says the growing incidence... There is no data before it to show that the incidence is growing. Has there been a historical change from the incidence to the 5 in 100,000? 

5. On page 1, line 32 mentioned peptides. It would be good to give a couple of examples. 

6. On page 2, line 64 brings up the topic of ZES but does not define what zollinger-ellison syndrome is. This should be done. 

7. I feel the discussion needs a little bit more. It is only 1 paragraph and mainly focus on the one meta-analysis. there are others published that might also need to be brought into the discussion. 

The english is well done and I have no real concerns.

Author Response

Attached below

Round 2

Reviewer 2 Report

Thank you to the authors for effectively answering my questions. No other comments required as I think the manuscript is fine for publication.